# Synergistic Antibiotic Activity of Ricini Semen Extract with Oxacillin against Methicillin-Resistant *Staphylococcus aureus*

**DOI:** 10.3390/antibiotics12020340

**Published:** 2023-02-06

**Authors:** Minjun Kim, Yena Seo, Seon-Gyeong Kim, Yedam Choi, Hyun Jung Kim, Tae-Jong Kim

**Affiliations:** 1Department of Forest Products and Biotechnology, College of Science and Technology, Kookmin University, 77 Jeongneung-ro, Seongbuk-gu, Seoul 02707, Republic of Korea; 2Department of Applied Chemistry, College of Science and Technology, Kookmin University, 77 Jeongneung-ro, Seongbuk-gu, Seoul 02707, Republic of Korea

**Keywords:** membrane fluidity, methicillin-resistant *Staphylococcus aureus*, oxacillin, Ricini Semen extract, synergistic antibiotic activity

## Abstract

Resistant bacteria are emerging as a critical problem in the treatment of bacterial infections by neutralizing antibiotic activity. The development of new traditional mechanisms of antibiotics is not the optimal solution. A more reasonable approach may be to use relatively safe, plant-based compounds in combination with conventional antibiotics in an effort to increase their efficacy or restore their activity against resistant bacteria. We present our study of mixing Ricini Semen extract, or its constituent fatty acids, with oxacillin and testing the effects of each on the growth of methicillin-resistant *Staphylococcus aureus.* Changes in the cell membrane fluidity of methicillin-resistant *S*. *aureus* were found to be a major component of the mechanism of synergistic antibiotic activity of Ricini Semen extract and its constituent fatty acids. In our model, changes in cellular membrane fluidity disrupted the normal function of bacterial signaling membrane proteins BlaR1 and MecR1, which are known to detect oxacillin, and resulted in the incomplete expression of penicillin-binding proteins 2a and β-lactamase. Utilizing the mechanism presented in this study presents the possibility of developing a method for treating antibiotic-resistant bacteria using traditional antibiotics with plant-based compounds.

## 1. Introduction

One of the most critical threats to human health is the emergence of antibiotic-resistant bacteria [1,2]. In 2019, 4.95 million deaths were related to antimicrobial resistance based on predictive statistical models [3]. According to a study of pathogenic strains in the Korean-Global AMR Surveillance System, from May 2016 through April 2017, *Staphylococcus aureus* was the third most common species of pathogens found in blood infections detected during hospitalization, accounting for 16.6%, of which more than half were methicillin-resistant *S*. *aureus* (MRSA) [4]. *S*. *aureus* is a skin bacterium that destroys the skin barrier and stimulates immunity, causing various skin diseases, including atopic dermatitis [5,6]. MRSA and vancomycin-resistant *S*. *aureus* are both well-known and clinically important antibiotic-resistant bacteria [7].

MRSA presents a clinical challenge to multidrug resistance (MDR) as well as resistance to methicillin [8]. MRSA is resistant to other antibiotics, including those impacting bacterial protein synthesis, which includes chloramphenicol, tetracycline, and erythromycin [9,10] and those impacting bacterial DNA synthesis, such as ciprofloxacin [9,10]. Currently, administering broad-spectrum antibiotics is the standard of care used to cure MDR–MRSA infections [11]. However, the development of new antibiotics is not a sustainable solution to the problems of antibiotic-resistant bacteria. The introduction of new drugs for drug-resistant bacteria very quickly leads to the emergence of new strains of resistant bacteria [12,13].

Synergistic agents that increase the antibiotic activity of traditional antibiotics are one of the most promising options for successfully treating resistant bacterial infections [14]. For synergistic activities, a mixture of antibiotics with different mechanisms or a mixture of antibiotics with antimicrobial peptides have been used [15,16,17]. Natural substances, such as epigallocatechin gallate and baicalein, have also been found to increase antibiotic activity [18,19].

*Ricinus communis*, the castor oil plant, is a species of flowering plant that produces an oily seed, Ricini Semen [20]. Castor oil obtained from Ricini Semen is used in lubricants, food additives, and cosmetics [21,22,23]. According to previous studies, it has been suggested that castor oil has antimicrobial activity against *S*. *aureus* and *Escherichia coli* [24,25]. The antimicrobial mechanisms of Ricini Semen extract and the mechanism of its synergism with antibiotics have yet to be defined.

Ricinoleic acid, an unsaturated omega-9 fatty acid containing a hydroxyl group, accounts for approximately 85% of castor oil [26,27,28]. Ricinoleic acid has been reported to have antimicrobial activity against *S*. *aureus* [29]. Palmitic acid is a saturated fatty acid with 16 carbons accounting for about 2.8–4.3% of Ricini Semen ethanol extract [30]. Palmitic acid has been reported to have antimicrobial activity against MRSA, which increases when combined with both oxacillin and the surfactant Span 85 [31]. Stearic acid is a saturated fatty acid with 18 carbons and accounts for 2.8–3.3% of Ricini Semen ethanol extract. [30]. Stearic acid has not been found to inhibit the growth of *S. aureus* when treated at 20 µg/mL [32]. Linoleic acid and oleic acid are also present in Ricini Semen ethanol extract [30].

Oxacillin is a β-lactam antibiotic and a derivative of penicillin with high antibiotic activity against *Staphylococcus* spp. and other Gram-positive bacteria [33,34]. The mechanism of resistance to β-lactam antibiotics in *S. aureus* is that bacterial cellular membrane sensor proteins, MecR1/BlaR1, detect the antibiotic and deactivate the repressor proteins MecI/BlaI [35]. The released promoter expresses MecA, a penicillin-binding protein 2a (PBP2a), and BlaZ, a β-lactamase [35,36,37]. In bacterial strains without the *mecR1*-*mecI* system, a *blaR1*-*blaI* system with a high degree of homology with *mecR1*-*mecI,* takes over and serves to regulate the expression of *mecA* [38].

*mecA* is present in a mobile genetic element, staphylococcal cassette chromosome *mec* (SCC*mec*) [39]. Currently, a total of 13 SCC*mec* types are classified [40]. According to the MRSA survey of Korean hospitals in the years from 2017 to 2019, 47.5% of MRSA isolated from infected patients were SCC*mec* type II, and 50.1% were SCC*mec* type IV [41].

Extracts having synergistic antibiotic activity with oxacillin against MRSA were identified within the 397-edible plant extract library (data not shown). Among them, we found that Ricini Semen extract and its chemical components, ricinoleic acid, palmitic acid, and stearic acid, conferred synergistic activity with oxacillin against two of the SCC*mec* types of MRSA. This study also provides data regarding the possible biological mechanisms of Ricini Semen that inhibit MRSA growth when used in combination with oxacillin.

## 2. Results and Discussion

### 2.1. Characterization of Two MRSA Strains: SCCmec Type and Minimum Inhibitory Concentration

To confirm the SCC*mec* type of the MRSA CCARM 3807 and MRSA CCARM 3820 strains used in this study, we verified the presence of SCC*mec* in each strain (Appendix A). PCR confirmed that the MRSA CCARM 3807 strain was the SCC*mec* type II having *mecA*, *ccrA2*-*ccrB2*, and *mecA*-*mecI* and that the MRSA CCARM 3820 strain was SCC*mec* type IV having *mecA*, *ccrA2*-*ccrB*, and *mecA*-IS1272 instead of *mecA*-*mecI.*

The antibiotic MIC was evaluated to determine the concentration of oxacillin to be used to test for synergistic inhibitory activity with the test compounds (Table 1). Resistance of MRSA CCARM 3807 to oxacillin and Ricini Semen extract was higher than that of MRSA CCARM 3820. Whereas resistance to ricinoleic acid was the same in both strains, resistance to the four fatty acids, linoleic acid, oleic acid, palmitic acid, and stearic acid, could not be compared within the tested concentration ranges.

According to the National Committee for Clinical Laboratory Standards, *S. aureus* can be identified as MRSA when the MIC for oxacillin is 4 mg/L or more [42], so our study confirmed that both strains were MRSA. Palmitic acid and stearic acid both had MICs of higher than 500 µM against *S. aureus* in the previous study by Parsons et al. [43], which is consistent with our results. Oleic acid was also reported to have an MIC of 1250 mg/L for MRSA Rosenbach (ATCC BAA-1683) [44] as in this study. The MIC of linoleic acid was 312.5 mg/L on MRSA Rosenbach [44], but in our experiment, it was measured to be higher than 512 mg/L.

### 2.2. Analysis of the Chemical Composition of the Ricini Semen Extract by Gas Chromatography

The fatty acid contents of the Ricini Semen extract were analyzed using gas chromatography. The contents of ricinoleic acid, linoleic, oleic acid, palmitic acid, and stearic acid in the Ricini Semen extract were 85.8%, 4.7%, 3.9%, 3.9%, and 1.6%, respectively. The fatty acid contents of the Ricini Semen ethanol extract in a previous study were 69–76% ricinoleic acid, 9–10% linoleic acid, 7–9% oleic acid, 3–4% palmitic acid, and 3% stearic acid [30]. Compared to this previous study, our Ricini Semen extract contained fewer linoleic, oleic, and stearic acids.

### 2.3. Effects on the Cell Growth

The effect of each compound on cell growth was analyzed at a concentration of no more than 50% of the MIC (Figure 1). In the case of MRSA CCARM 3807, oxacillin, Ricini Semen extract, and ricinoleic acid inhibited bacterial cell growth by 10.4%, 19.5%, and 12.0%, respectively (Figure 1A). The combination of oxacillin and Ricini Semen extract or the combination of oxacillin and ricinoleic acid inhibited cell growth by 69.9% and 49.4%, respectively. The growth inhibitory activity of these two combinations was more than twice the growth inhibitory activity of the individual compounds, confirming its synergistic activity.

Palmitic acid and stearic acid also showed similar synergistic growth inhibitory activity (Figure 1B). The growth inhibition by oxacillin, palmitic acid, and stearic acid after 24 h was 3.0%, −8.4%, and −14.6%. In comparison, the growth inhibition was 17.3% by oxacillin combined with palmitic acid and 9.2% by the combination of oxacillin and stearic acid.

Cultures of MRSA CCARM 3820 were tested with oxacillin, Ricini Semen extract, and ricinoleic acid, and we found that cell growth was inhibited by 13.9%, 9.8%, and 27.5%, respectively (Figure 1C). The combination of oxacillin and Ricini Semen extract inhibited cell growth by 26.3%, and the combination of oxacillin and ricinoleic acid inhibited cell growth by 76.4%. 

Palmitic acid and stearic acid also showed synergistic activity with oxacillin (Figure 1D). Oxacillin, palmitic acid, and stearic acid inhibited cell growth by 23.6%, −11.0%, and −24.7%, respectively. The combination of oxacillin and palmitic acid inhibited cell growth by 70.2%, and stearic acid and oxacillin combined exposure inhibited growth by 86.7%.

Our results demonstrate that the antibiotic activity of oxacillin against two different MRSA strains was significantly increased through the addition of Ricini Semen extract and its fatty acid components. 

### 2.4. Synergistic Activity of Oxacillin on MRSAs

To evaluate the synergistic activity accurately, we analyzed the number of CFU of the 24-h culture, depicted in Figure 1 (Figure 2). The combination of oxacillin and the Ricini Semen extract increased the inhibitory activity 3.6 and 5.3 times compared to individual treatments of oxacillin or Ricini Semen extract, respectively, and when oxacillin and ricinoleic acid were combined, the inhibitory activity increased 2.9 and 36.4 times compared to the individual treatments of oxacillin and ricinoleic acid, respectively (Figure 2A).

Palmitic acid or stearic acid individual treatments did not inhibit the growth of MRSA CCARM 3807 after 24 hrs. However, in combination with oxacillin, cell growth was significantly inhibited by 2.5 and 2.3 times, respectively, compared to oxacillin alone (Figure 2B).

The MRSA CCARM 3820 strain CFUs that were treated with the combination of oxacillin and the Ricini Semen extract were reduced by 67.6 times compared to the individual oxacillin treatment and by 4.9 times compared to the Ricini Semen extract treatment. The combination treatment of oxacillin and ricinoleic acid reduced CFU by 77.6 times compared to individual oxacillin treatment and by 2.0 times when compared to ricinoleic acid alone (Figure 2C).

As with MRSA CCARM 3807, treatment with palmitic acid and stearic acid did not inhibit the growth of MRSA CCARM 3820. The cell growth inhibitory activity was 2.7 times stronger with the combination of oxacillin and palmitic acid and 2.8 times stronger with the combination of oxacillin and stearic acid than with oxacillin alone (Figure 2D).

All tested combination treatments inhibited cell growth more than oxacillin alone or any of the tested substances alone. This result verifies synergistic activity against both strains of MRSA in all tested combination treatments.

Several studies have previously reported that many compounds have been shown to increase the antibiotic activity of oxacillin. Thioxanthone showed synergistic activity in combination with ampicillin and oxacillin against MRSA in a model that binds to the allosteric domain of PBP2a [45]. Abietic acid and oxacillin have been reported to have synergistic activity against *S*. *pseudintermedius* and to significantly reduce the expression of *mecA*, *mecR1*, and *mecI* [46]. The ethanol extract of *Hwangheuk-san* was shown to synergistically increase oxacillin antibiotic activity against MRSA [47]. One of the flavanonols, rhamnoside, has been demonstrated to have synergism with ampicillin, a β-lactam antibiotic similar to oxacillin, against MRSA [48].

### 2.5. Changes in Cellular Membrane Fluidity by Ricini Semen Extract and Its Component Fatty Acids

The major components of Ricini Semen extract are fatty acids, and the primary target of fatty acids on the bacterium is speculated to be adverse changes in their cellular membrane [49]. Changes in cellular membrane fluidity by Ricini Semen extract and its component fatty acid were measured using laurdan (Figure 3). Laurdan provides a change of cellular membrane fluidity by entering the cell membrane bilayer and changing its emission wavelength according to membrane fluidity [50,51]. Laurdan GP, depicted in Figure 3, decreases when cellular membrane fluidity increases and increases when its fluidity decreases [52].

Ricini Semen extract, palmitic acid, and stearic acid decreased cellular membrane fluidity in both strains of MRSA. In comparison, ricinoleic acid, which constitutes more than 70% of the Ricini Semen extract, increased cellular membrane fluidity in both MRSA strains. A previous study [53] reported that ursolic acid and oleanolic acid both showed synergistic activity with ampicillin against MRSA. These two compounds reduced the cellular membrane fluidity of MRSA and induced the delocalization of PBP2 [53,54]. Compound C18, one of the cecropin-4-derived peptides, also demonstrated synergism with oxacillin against MRSA and increased the cellular membrane fluidity of MRSA [55]. The changes in cellular membrane fluidity seen in this study and in the results of previous studies suggest that Ricini Semen extract and its component fatty acids have synergistic activity with oxacillin through the changes in cellular membrane fluidity.

### 2.6. Changes in Antibiotic Resistance-Related Gene Expression by Ricini Semen Extract and Its Component Fatty Acids

PBP2a, a penicillin-binding protein synthesized from *mecA*, confers resistance against β-lactam antibiotics, including methicillin and oxacillin, to *S*. *aureus* [56]. MRSA CCARM 3807 was found to have the SCC*mec* type II element. In one model [57,58], a signaling membrane receptor MecR1 recognizes oxacillin and becomes activated as a result. The activated MecR1 degrades the repressor MecI, and the expression of *mecA* is subsequently increased. At the same time, a different signaling membrane receptor, BlaR1, with a similar function to MecR1, also recognizes oxacillin and becomes activated. The activated BlaR1 degrades its own repressor BlaI, and the expression of *blaZ,* the gene for β-lactamase (PC1), is significantly upregulated [57]. 

The MRSA CCARM 3820 had an SCC*mec* type IV element with no *mecI* and a truncated *mecR1* [58]. Because this strain does not have a functional MecI and MecR1, only BlaI and BlaR1 regulate the expression of both *mecA* and *blaZ* [37,59]. Oxacillin is detected by BlaR1 on the cellular membrane. The activated BlaR1 then degrades the repressor BlaI. By the suppression of BlaI, the expressions of *blaZ* for PC1 and *mecA* for PBP2a were both significantly upregulated [57].

Expression level changes of *mecA*, *mecR1*, and *blaR1* by Ricini Semen extract and ricinoleic, palmitic, and stearic acids were analyzed using RT-PCR (Figure 4). Oxacillin was found to increase the expression of all three genes, *mecA*, *mecR1*, and *blaR1*, but Ricini Semen extract and ricinoleic acid decreased the expression of all three genes in the MRSA CCARM 3807 strain. When Ricini Semen extract or ricinoleic acid was additionally treated with oxacillin, the expression levels of all three genes were lower than with oxacillin alone (Figure 4A).

In the same experiment, using palmitic acid and stearic acid (Figure 4B), oxacillin upregulated all three genes, *mecA*, *mecR1*, and *blaR1,* but palmitic acid and stearic acid downregulated them. The combination of palmitic acid or stearic acid with oxacillin downregulated gene expression compared to oxacillin alone.

Since *mecR1* was not found to be present in MRSA CCARM 3820, only *mecA* and *blaR1* gene expression changes were measured (Figure 4C,D). Oxacillin increased the expression of these two genes, but Ricini Semen extract and ricinoleic acid, palmitic acid, and stearic acid decreased their expression. When oxacillin was combined with either Ricini Semen extract, ricinoleic acid, palmitic acid, or stearic acid, the expression of both genes decreased compared to oxacillin alone and compared to the Ricini Semen extract, ricinoleic acid, palmitic acid, and stearic acid, but not as much as in untreated cells.

One of the mechanisms that increases resistance to oxacillin in *S*. *aureus* is the increased expression of *mecA*, *mecR1*, and *blaR1*. Therefore, the downregulation of *mecR1* and *blaR1* by the Ricini Semen extract and ricinoleic acid, palmitic acid, and stearic acid all reduced the degradation of MecI and BlaI. The repressors, MecI and BlaI, repressed *mecA* expression and increased the bacterial sensitivity to oxacillin.

A previous study [60] reported that thioridazine acts on the cellular membrane, causing conformational changes in the membrane proteins, BlaR1 and MecR1. The denatured membrane proteins do not sufficiently degrade the repressors, so the expression of *mecA* and *blaZ* is suppressed even in the presence of oxacillin. Decreased amounts of PBP2a and PC1 increase the sensitivity of MRSA to β-lactam antibiotics.

Ricini Semen extract, ricinoleic acid, palmitic acid, and stearic acid all affected cellular membrane fluidity and may cause membrane proteins BlaR1 and MecR1 to malfunction. In a previous study [60], cellular membrane fluidity changes were not measured or presented as a model, but in this study, the changes in cellular membrane fluidity of MRSA by Ricini Semen extract, ricinoleic acid, palmitic acid, and stearic acid were measured. Ricini Semen extract, palmitic acid, and stearic acid all decreased cellular membrane fluidity, but ricinoleic acid, which constitutes more than 70% of Ricini Semen extract, increased cellular membrane fluidity. This suggests that the effect of Ricini Semen extract on cell membrane fluidity is determined by palmitic acid and stearic acid, which constitute only 4.6% of the extract, rather than ricinoleic acid, which is the main component. Ricinoleic acid synergistically increased the antibiotic activity of oxacillin regardless of the increase or decrease in cell membrane fluidity in our study. This means that membrane fluidity in untreated cells provides the optimal condition for the normal function of BlaR1 and MecR1, and its function diminishes when cellular membrane fluidity changes significantly.

## 3. Materials and Methods

### 3.1. Bacterial Strains and Culture Conditions

MRSA CCARM 3807 [61] and CCARM 3820 were purchased from the Culture Collection of Antimicrobial Resistance Microbes at Seoul Women’s University (Seoul, Republic of Korea). The strains were stored in 25% glycerol (catalog number: 4066-4400, Daejung Chemical & Metals Co., Ltd., Siheung, Republic of Korea) at −80 °C.

Both strains were cultured in Tryptic Soy Broth (TSB, catalog number: 211825, Becton Dickinson Korea Co., Ltd., Seoul, Republic of Korea) and plated on Tryptic Soy Agar (TSA) plates made by adding 1.5% Bacto-Agar (catalog number: 214010, Becton Dickinson Korea) to TSB.

Saline was prepared by adding 0.85% (*w/v*) of sodium chloride (catalog number: S0476, Samchun Chemicals Co., Ltd., Seoul, Republic of Korea) to distilled water. All solutions were sterilized at 121 °C for 20 min. The strains were cultured at 37 °C with a shaking speed of 250 rpm for liquid cultures.

### 3.2. Chemicals

Oxacillin sodium salt (catalog number: sc-224180B, Santa Cruz Biotechnology Inc., Dallas, TX, USA) was dissolved in distilled water. Ricinoleic acid (catalog number: R7257, Sigma-Aldrich Inc., Seoul, Republic of Korea), palmitic acid (catalog number: 129702500, Acros Organics, ThermoFisher Scientific Korea Ltd., Seoul, Republic of Korea), stearic acid (catalog number: S0163, Tokyo Chemical Industry Co., Ltd., Tokyo, Japan), linoleic acid (catalog number: L07949, ThermoFisher Scientific Korea), and oleic acid (catalog number: O0180, Tokyo Chemical Industry) were dissolved in ethanol (catalog number: 000E0243, Samchun Chemicals). Laurdan (6-Dodecanoyl-2-Dimethylaminonaphthalene, catalog number: D250) was purchased from Invitrogen (ThermoFisher Scientific Korea). Phosphate-buffered saline (PBS, catalog number: P5493-1L) was purchased from Sigma-Aldrich Inc.

### 3.3. Preparation of Ricini Semen Ethanol Extract

Ricini Semen ethanol extract was prepared according to a previous study [62] with some modifications. Ricini Semen, purchased from Jiundang Oriental Pharmacy (Seoul, Republic of Korea), was crushed to a size of 5 mm or less. Thirty grams of the crushed seed was mixed with 300 mL of ethanol (catalog number: 000E0219, Samchun Chemicals) in a 500 mL round flask and incubated at 50 °C for 3 h while shaking at 30 min intervals. After the incubation, the solids were removed by filtration using Whatman qualitative filter paper Grade 1 (catalog number: 1002 110, Cytiva, Sigma-Aldrich). The filtered extract was then concentrated using a rotary evaporator (RV-10, IKA Korea Ltd., Seoul, Republic of Korea), and the residual ethanol was removed using a freeze dryer (FDU-2110, EYELA, SunilEyela Co., Ltd., Seongnam, Republic of Korea). The solid weight after lyophilization was 3.29 g. The extracted oil was stored at −80 °C.

### 3.4. Identification of SCCmec Type of MRSA Strains

The SCC*mec* type of MRSA strain was confirmed by polymerase chain reaction (PCR) according to a previous study [63]. The primers were mA1 (5′-TGCTATCCACCCTCAAACAGG-3′) and mA2 (5′-AACGTTGTAACCACCCCAAGA-3′) for *mecA*, α2 (5′-TAAAGGCATCAATGCACAAACACT-3′) and βc (5′-ATTGCCTTGATAATAGCCTCT-3′) for *ccrA2*-*ccrB2*, mA7 (5′-ATATACCAAACCCGACAACTACA-3′) and mI6 (5′-CATAACTTCCCATTCTGCAGATG-3′) for *mecA*-*mecI*, and mA7 and IS7 (5′-ATGCTTAATGATAGCATCCGAATG-3′) for *mecA*-IS1272. Genomic DNA was extracted using the MiniBEST Bacteria Genomic DNA Extraction Kit Ver.3.0 (catalog number: 9763A, Takara Korea Biomedical Inc., Seoul, Republic of Korea). The PCR reaction mixture consisted of 0.8 µL of chromosomal DNA, 0.8 µL of dNTP (each 10 mM), 4 µL of ×10 pfu buffer, 0.4 µL of forward primer (50 pmol/µL), 0.4 µL of reverse primer (50 pmol/µL), 0.4 µL of pfu polymerase (catalog number: SPD95-E500, SolGent Co., Ltd., Deajeon, Republic of Korea), and 33.2 µL of distilled water. PCR conditions for *mecA* and *ccrA2*-*ccrB* were one cycle of initial denaturation at 94 °C for 2 min, 30 cycles of denaturation at 94 °C for 2 min, annealing at 57 °C for 1 min and extension at 72 °C for 2 min, and one cycle of final elongation at 72 °C for 2 min [63].

For *mecA*-*mecI* and *mecA*-IS1272, all PCR conditions were the same as those described above, except the annealing temperature was increased from 57 °C to 60 °C [63]. The PCR product was stained with Dyne Loading STAR (catalog number: A750, Dyne Bio Inc., Seongnam, Republic of Korea) after DNA gel electrophoresis with 1.5% agarose gel.

### 3.5. Measuring the Minimal Inhibition Concentration of the MRSA Strains

MRSA CCARM 3807 and MRSA CCARM 3820 were plated on TSB agar. The plates were then incubated at 37 °C for 24 h, after which the bacteria were used to inoculate 5 mL of TSB and precultured at 37 °C for 24 h with 250 rpm continuous shaking.

The precultured cells with Abs_600_ = 0.05 of the final cell densities were subcultured into another 5 mL of TSB medium to which each chemical or extract was added. These cultures were incubated at 37 °C for 24 h with 250 rpm continuous shaking, after which the cell density by Abs_600_ was measured. Each experiment was repeated twice over three experiments total for each set of conditions. The minimum concentration at which cells did not grow in all three cultures was calculated as the minimum inhibitory concentration (MIC).

### 3.6. Analysis of the Chemical Composition of the Ricini Semen Extract by Gas Chromatography

The Ricini Semen extract was analyzed by gas chromatography for the quantification of each component, including linoleic acid, oleic acid, palmitic acid, ricinoleic acid, and stearic acid. The samples were pretreated according to the protocol used in a previous study [26] with few modifications. Samples were incubated at 80 °C for 30 min with 4 mL of 3 N methanolic HCl in a water bath for acid-catalyzed transesterification. After cooling to room temperature, samples were vortexed for 1 min with 2 mL of hexane and 1 mL of 0.85% (*w/v*) sodium chloride. After centrifugation at 699× *g* for 1 min and subsequent filtration with syringeless filters 0.20 µm (catalog number: MV32ANPPT002TC01, GVS Korea Ltd., Namyangjoo, Republic of Korea), the hexane layer was analyzed.

The fatty acids in the Ricini Semen extract were analyzed by gas chromatography (model: 7890B, Agilent Technologies, Inc., Wilmington, DE, USA) with a flame ionization detector and with the use of a column (HP-INNOWax 19091N-133, Agilent Technologies). The oven conditions were held at 120 °C for 1 min, after which the temperature was increased at a rate of 10 °C per minute to 240 °C, and this temperature was maintained for 10 min. The gas flow was set at 1.8 mL/min, and 1 µL of each sample was analyzed in split mode 1:50. The mobile phase gas was N_2_, the injector temperature was set at 270 °C, the detector temperature was set at 280 °C, the H_2_ flow rate was set at 30 mL/min, and the airflow rate was set at 300 mL/min [26].

### 3.7. Measuring the Growth Curve

Bacterial cells were cultured as described in the section “Measuring the minimal inhibition concentration of the MRSA strains” up to the preculture step. The precultured cells were inoculated into 20 mL aliquots of TSB medium to which each chemical or extract was added. This culture was then tested so that the bacterial concentration at Abs_600_ = 0.05. The concentrations of compounds and extracts used to measure the cell growth curve are specified in corresponding sections. The cultures were incubated at 37 °C for 24 h with 250 rpm of continuous shaking. The concentration of bacterial cells was measured by absorbance at 600 nm at hourly intervals. The experiment was repeated three times, and the mean and standard deviations were calculated.

### 3.8. Measuring Colony-Forming Units and Evaluating Synergistic Growth Inhibition

Colony-forming units (CFU) were measured to estimate the number of living bacterial cells after culture. In the previous growth curve experiment, a 24-h culture sample was diluted with saline and spread on TSA plates. After incubation at 37 °C for 24 h, the number of bacterial colonies on the TSA plates was counted. The experiment was repeated three times, and the mean and standard deviations were calculated.

The synergistic antibiotic activity was determined by using the CFU value instead of the absorbance of the culture medium in a previous paper [64]. The relative growth inhibition was calculated by the following formula: (CFU_Control_ − CFU_Sample_)/CFU_Control_ × 100. CFU_Control_ is the CFU value of cell culture without any testing extract or compound. Compared with the relative growth inhibition of each sample, if the relative growth inhibition by mixing two samples was more than doubled, it was designated as synergistic activity.

### 3.9. Gene Expression Analysis using Real-Time Polymerase Chain Reaction

The cell culture and material treatment conditions for this portion of our study are the same as in our growth curve experiment. After the main culture for 3 h, cells were broken by TissueLyser LT (Qiagen Korea Ltd., Seoul, Republic of Korea) [65], and RNA was extracted using an AccuPrep Bacterial RNA Extraction Kit (Bioneer Co., Daejeon, Republic of Korea) according to the manufacturer’s manual.

A total of 3 µL of the purified RNA template, 1 µL of forward primer (5 pmol/µL), 1 µL of reverse primer (5 pmol/µL), 1 µL of enzyme mix from the TOPreal One-step RT qPCR Kit (SYBR Green with low ROX) (Enzynomics Co., Ltd., Daejeon, Republic of Korea), 10 µL of reaction mix from the TOPreal One-step RT qPCR Kit, and 4 µL of water (RNase free) were mixed, and the mRNA concentration was analyzed using QuantStudio5 (ThermoFisher Scientific Korea) in preparation for the real-time polymerase chain reaction (RT-PCR).

The primer sequences were 16S rRNA forward (5′-TCCGGAATTATTGGGCGTAA-3′) [66], 16S rRNA reverse (5′-CCACTTTCCTCTTCTGCACTCA-3′) [66], *mecA* forward (5′-GTTAGATTGGGATCATAGCGTCATT-3′) [66,67], *mecA* reverse (5′-TGCCTAATCTCATATGTGTTCCTGTAT-3′) [66,67], *mecR1* forward (5′-AAGCACCGTTACTATCTGCACA-3′) [68], *mecR1* reverse (5′-GAGTAATTTTGGTCGAATGCC-3′) [68], *blaR1* forward (5′-CATGACAATGAAGTAGAAGC-3′) [69], and *blaR1* reverse (5′-CTTATGATTCCATGACATACG-3′) [69]. The cDNA synthesis of all genes by reverse transcription was performed at 50 °C for 30 min. The analysis conditions for RT-PCR for *mecA* were initial denaturation at 95 °C for 10 min followed by 40 cycles of denaturation at 95 °C for 15 sec, annealing at 60 °C for 1 min, and extension at 72 °C for 30 sec. Initial denaturation for *blaR1* was performed at 95 °C for 10 min, followed by 45 cycles of denaturation at 95 °C for 30 sec, annealing at 51 °C for 30 sec), and extension at 72 °C for 30 sec. Initial denaturation of *mecR1* was completed at 95 °C for 10 min, followed by 40 cycles of denaturation at 95 °C for 10 sec, annealing at 60 °C for 15 sec, and extension at 72 °C for 20 sec. The specificity of the PCR products was confirmed by the performance of a melting curve analysis between 60 °C and 95 °C. This experiment was performed in duplicate.

### 3.10. Analysis of Bacterial Cellular Membrane Fluidity Using Laurdan

Bacterial cellular membrane fluidity was measured according to the protocol presented in a previous study, with few modifications [70]. The bacterial culture and material treatment conditions were identical to those used in our growth curve experiment. We incubated the main bacterial cultures for 3 h, after which 10 µM laurdan was added, and the cells were then incubated for an additional 10 min. The cells were then washed with PBS three times, after which they were resuspended in PBS at the same concentration as before washing. A total of 200 µL of this cell culture suspension was dispensed into a black 96-well plate (catalog number: 30296, SPL Life Sciences Co., Ltd., Pocheon, Republic of Korea) and incubated for 1 h at room temperature. Fluorescence was measured with excitation at 350 nm and emission at 435 nm and 490 nm using Varioskan LUX multimode microplate reader (catalog number: VLBL00D0, ThermoFisher Scientific Korea). The cell membrane fluidity was calculated as laurdan generalized polarization (GP) = (*I*_435_ − *I*_490_)/(*I*_435_ + *I*_490_). The experiment was performed independently in triplicate, and the mean value and standard deviation value were calculated.

## 4. Conclusions

Ricini Semen extract and its constituent fatty acids synergistically enhance the antibiotic activity of oxacillin. The cellular membrane fluidity of *S. aureus* is changed by Ricini Semen extract and its constituent fatty acids. The altered fluidity in the proposed model diminishes the ability of membrane proteins BlaR1 and MecR1 to detect oxacillin in the SCC*mec* element, conferring resistance to oxacillin. Interfering with the degradation of BlaI and MecI by inactivating BlaR1 and MecR1, respectively, may prevent the expression of β-lactamase (BlaZ) and penicillin-binding proteins 2a (PBP2a), which are resistance mechanisms induced by oxacillin. Ricini Semen extract and its constituent fatty acids are suggested to enhance the activity of β-lactam antibiotics and restore their therapeutic activity on MRSA infections.

## Figures and Tables

**Figure 1 antibiotics-12-00340-f001:**
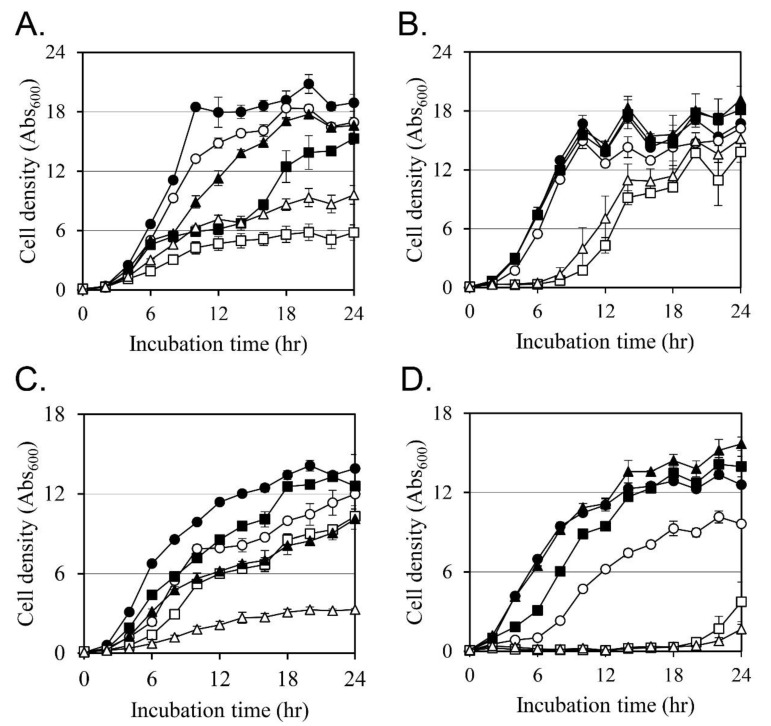
Effect on the growth of MRSA CCARM 3807 and MRSA CCARM 3820. In (**A**), cell growth of MRSA CCARM 3807 was tested without compounds as a control (●), with 64 mg/L of oxacillin (○), with 64 mg/L of Ricini Semen extract (■), with 64 mg/L of Ricini Semen extract combined with 64 mg/L of oxacillin (☐), with 32 mg/L of ricinoleic acid (▲), and with 32 mg/L of ricinoleic acid combined with 64 mg/L of oxacillin (△). In (**B**), cell growth of MRSA CCARM 3807 was tested without compounds as a control (●), with 64 mg/L of oxacillin (○), with 128 mg/L of palmitic acid (■), with 128 mg/L of palmitic acid combined with 64 mg/L of oxacillin (☐), with 128 mg/L of stearic acid (▲), and with 128 mg/L of stearic acid combined with 64 mg/L of oxacillin (△). In (**C**), cell growth of MRSA CCARM 3820 was tested without compounds as a control (●), with 4 mg/L of oxacillin (○), with 64 mg/L of Ricini Semen extract (■), with 64 mg/L of Ricini Semen extract combined with 4 mg/L of oxacillin (☐), with 64 mg/L of ricinoleic acid (▲), and with 64 mg/L of ricinoleic acid combined with 4 mg/L of oxacillin (△). In (**D**), cell growth of MRSA CCARM 3820 was tested without compounds as a control (●), with 4 mg/L of oxacillin (○), with 128 mg/L of palmitic acid (■), with 128 mg/L of palmitic acid in combination with 4 mg/L of oxacillin (☐), with 128 mg/L of stearic acid (▲), and with the combination of 128 mg/L of stearic acid and 4 mg/L of oxacillin (△). All components of this experiment were repeated three times, and the mean and standard deviations were calculated.

**Figure 2 antibiotics-12-00340-f002:**
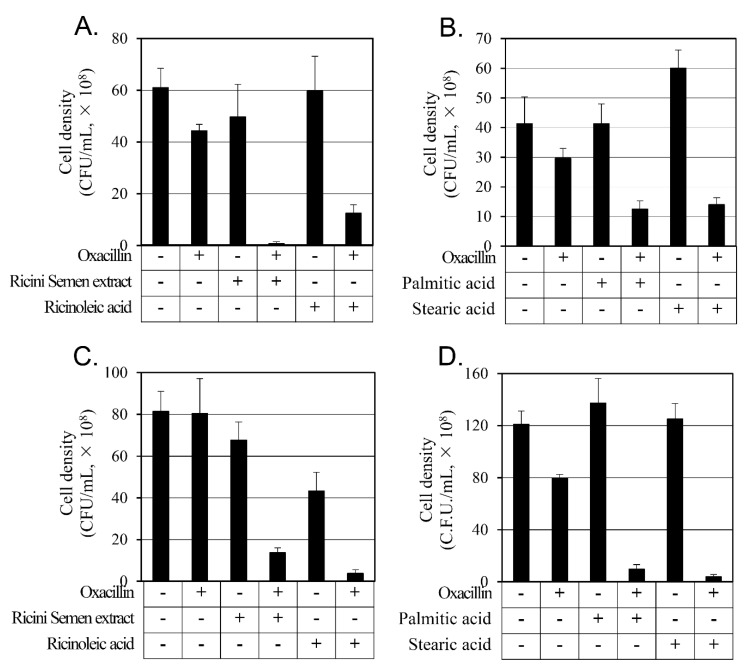
Evaluating the synergistic activity on *S. aureus*. To evaluate the synergistic activity, the cell density was measured in colony-forming units (CFU) after 24-h incubation. In (**A**), the cell density of MRSA CCARM 3807 was measured after 24 h incubation with either 64 mg/L of oxacillin, 64 mg/L of Ricini Semen extract, or 32 mg/L of ricinoleic acid. In (**B**), the cell density of MRSA CCARM 3807 was measured after 24 h incubation with either 64 mg/L of oxacillin, 128 mg/L of palmitic acid, or 128 mg/L of stearic acid. In (**C**), the cell density of MRSA CCARM 3820 was measured after 24 h incubation with either 4 mg/L of oxacillin, 64 mg/L of Ricini Semen extract, or 64 mg/L of ricinoleic acid. In (**D**), the cell density of MRSA CCARM 3820 was measured after 24 h incubation with either 4 mg/L of oxacillin, 128 mg/L of palmitic acid, or 128 mg/L of stearic acid. Each experiment was repeated three times, and the mean and standard deviations were calculated.

**Figure 3 antibiotics-12-00340-f003:**
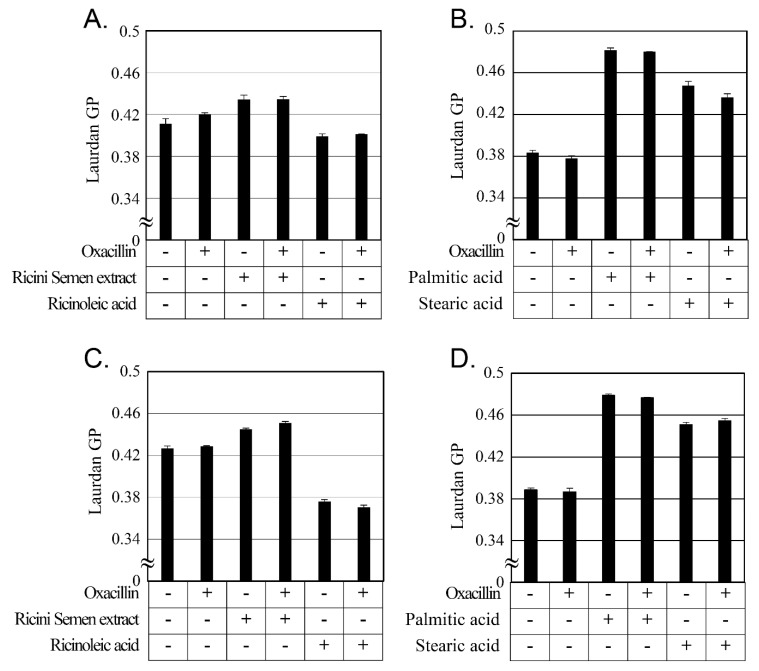
Cellular membrane fluidity changes induced by Ricini Semen extract and its constituent fatty acids. Laurdan generalized polarization (GP) decreases when cell membrane fluidity increases and increases when fluidity decreases. In (**A**), the membrane fluidity of MRSA CCARM 3807 was measured with 64 mg/L of oxacillin, 64 mg/L of Ricini Semen extract, or 32 mg/L of ricinoleic acid. In (**B**), the membrane fluidity of MRSA CCARM 3807 was measured with either 64 mg/L of oxacillin, 128 mg/L of palmitic acid, or 128 mg/L of stearic acid. In (**C**), the membrane fluidity of MRSA CCARM 3820 was measured with either 4 mg/L of oxacillin, 64 mg/L of the Ricini Semen extract, or 64 mg/L of ricinoleic acid. In (**D**), the membrane fluidity of MRSA CCARM 3820 was measured with either 4 mg/L of oxacillin, 128 mg/L of palmitic acid, or 128 mg/L of stearic acid. Each of these experiments was performed three times independently, and the mean and standard deviations were calculated.

**Figure 4 antibiotics-12-00340-f004:**
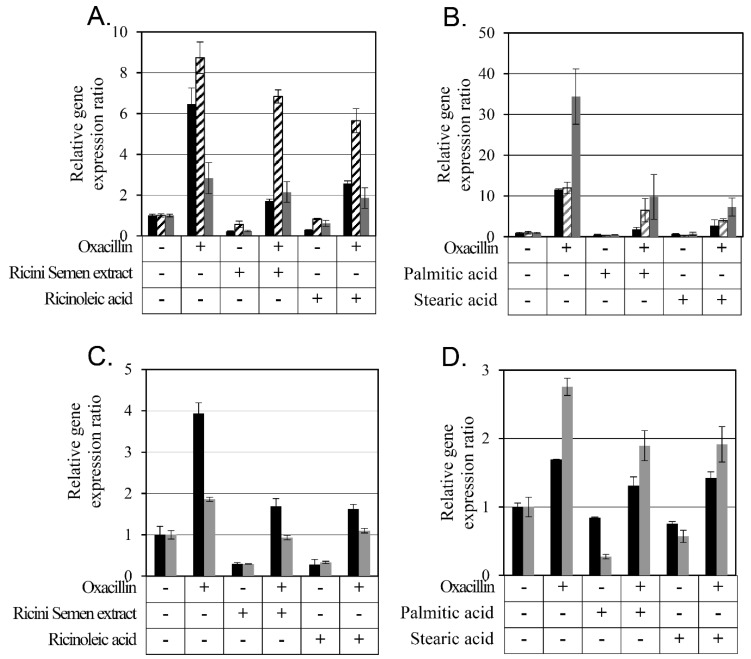
Gene expression change in SCC*mec* element induced by Ricini Semen extract and its constituent fatty acids. The relative expression levels of *mecA* (black bars), *mecR1* (slashed bars), and *blaR1* (gray bars) were calculated by comparing them with the expression level of 16S rRNA measured using RT-PCR. In (**A**), the relative gene expression of MRSA CCARM 3807 was measured after 3-h incubation with 64 mg/L of oxacillin, 64 mg/L of Ricini Semen extract, or 32 mg/L of ricinoleic acid. In (**B**), the relative gene expression of MRSA CCARM 3807 was measured after 3 h incubation with either 64 mg/L of oxacillin, 128 mg/L of palmitic acid, or 128 mg/L of stearic acid. In (**C**), the relative gene expression of MRSA CCARM 3820 was measured after 3 h incubation with either 4 mg/L of oxacillin, 64 mg/L of the Ricini Semen extract, or 64 mg/L of ricinoleic acid. In (**D**), the relative gene expression of MRSA CCARM 3820 was measured after 3 h incubation with either 4 mg/L of oxacillin, 128 mg/L of palmitic acid, or 128 mg/L of stearic acid. Each of these experiments was performed in duplicate.

**Table 1 antibiotics-12-00340-t001:** Minimal inhibition concentration of the two MRSA strains.

Strain	MIC (mg/L)						
	Oxacillin	Ricini Semen Extract	Ricinoleic Acid	Linoleic Acid	Oleic Acid	Palmitic Acid	Stearic Acid
CCARM 3807	1024	>1024	256	>512	>512	>256	>256
CCARM 3820	32	256	256	>512	>512	>256	>256

## Data Availability

The data presented in this study are available on request from the corresponding author. The data are not publicly available due to intellectual property protection.

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
