# Peer review of "Synergistic Antibiotic Activity of Ricini Semen Extract with Oxacillin against Methicillin-Resistant Staphylococcus aureus"

_antibiotics, 2023, doi:10.3390/antibiotics12020340_

Round 1

Reviewer 1 Report

In the manuscript titled "Synergistic Antibiotic Activity of Ricini Semen Extract with Oxacillin against Methicillin-resistant Staphylococcus aureus" the authors mixed Ricini Semen extract or its constituent fatty acids with oxacillin and tested the effects of each on the growth of methicillin-resistant Staphylococcus aureus. The manuscript is interesting and well written. However, I have comments for the authors to address before the paper is recommended for publication.

1. Line 117-123: In the results and discussion section, I suggest that authors add a new subsection titled “Analysis of the Chemical Composition of the Ricini Semen Extract by Gas Chromatography” to show the results of extract analysis by Gas Chromatography in a separate topic.

2. Line 303-312: In subsection “3.3. Preparation of Ricini Semen Ethanol Extract”, the authors reported that 30 g of crushed seed and 300 mL of ethanol were used to obtain the extract, however, the final yield of the crude extract was not informed. Please, insert this information.

3. Page 19: The presentation of Figure 1 could be improved. As it stands, it is difficult for the reader to visualize the cell growth curves corresponding to each evaluated product, bearing in mind that some curves overlap. Perhaps it would be interesting to substitute other styles of more didactic graphics to facilitate the interpretation of the results.

Author Response

  1. Line 117-123: In the results and discussion section, I suggest that authors add a new subsection titled “Analysis of the Chemical Composition of the Ricini Semen Extract by Gas Chromatography” to show the results of extract analysis by Gas Chromatography in a separate topic.

Response: Line 132-139. A new section was created.

2.2. Analysis of the Chemical Composition of the Ricini Semen Extract by Gas Chromatography

  1. Line 303-312: In subsection “3.3. Preparation of Ricini Semen Ethanol Extract”, the authors reported that 30 g of crushed seed and 300 mL of ethanol were used to obtain the extract, however, the final yield of the crude extract was not informed. Please, insert this information.

Response: Line 314. Added the information.

“The solid weight after lyophilization was 3.29 g.”

  1. Page 19: The presentation of Figure 1 could be improved. As it stands, it is difficult for the reader to visualize the cell growth curves corresponding to each evaluated product, bearing in mind that some curves overlap. Perhaps it would be interesting to substitute other styles of more didactic graphics to facilitate the interpretation of the results.

Response: It seems that figure 1 is showing a lot of results and is not readable. To improve readability, the data was changed and displayed at intervals of 1 hour to 2 hours.

Reviewer 2 Report

1- Bacterial Strains and Culture Conditions: refernce?

2-Preparation of Ricini Semen Ethanol Extract: refernce?

3-The analysis of the chemical compounds of the extract should be determined? (GC/MS)

4-use some other research in introduction and discussion section:

Prevalence, Antibiotic Resistance, Toxin-Typing and Genotyping of Clostridium perfringens in Raw Beef Meats Obtained from Qazvin City, Iran

5-Replace old REFERNCES.

Author Response

  1. Bacterial Strains and Culture Conditions: reference?’

Response: References to the MRSA CCARM 3807 strain are shown in [61]. However, as there are no previous studies on the MRSA CCARM 3820 strain, references cannot be added. The culture conditions for the strains in this study are described in two sections, one for “3.5. Measuring the Minimal Inhibition Concentration of the MRSA Strains” in line 338-348 and the other for “3.7. Described in the Measuring the Growth Curve” in line 368-377.

  1. Yadav, M.K.; Chae, S.-W.; Im, G.J.; Chung, J.-W.; Song, J.-J. Eugenol: A phyto-compound effective against methicillin-resistant and methicillin-sensitive Staphylococcus aureus clinical strain biofilms. PLoS One 2015, 10, e0119564.

  1. Preparation of Ricini Semen Ethanol Extract: reference?

Response: Added “Ricini Semen ethanol extract was prepared according to a previous study [62] with some modifications.” in line 306-307. And a reference for the basic extract procedures was added in reference 61.

  1. Ham, Y.; Kim, T.-J. Plant extracts inhibiting biofilm formation by Streptococcus mutans without antibiotic activity. Journal of the Korean Wood Science and Technology 2018, 46, 692-702.

  1. The analysis of the chemical compounds of the extract should be determined? (GC/MS)

Response: The chemical compounds of the extract were determined by Gas Chromatography (Section “3.6. Analysis of the Chemical Composition of the Ricini Semen Extract by Gas Chromatography” in line 349-367.). The analyzed results were described in a new section “2.2. Analysis of the Chemical Composition of the Ricini Semen Extract by Gas Chromatography” in line 132-139.  

  1. Use some other research in introduction and discussion section:

Prevalence, Antibiotic Resistance, Toxin-Typing and Genotyping of Clostridium perfringens in Raw Beef Meats Obtained from Qazvin City, Iran

Response: Added the recommended reference in line 49.

  1. Hassani, S.; Pakbin, B.; Brück, W.M.; Mahmoudi, R.; Mousavi, S. Prevalence, antibiotic resistance, toxin-typing and genotyping of Clostridium perfringens in raw beef meats obtained from Qazvin city, Iran. Antibiotics 2022, 11, 340.

  1. Replace old REFERENCES

Response: Reference 2 was changed from 2016 to 2022.

References 33 in 1962 and 34 in 1964 are the oldest but most relevant references.

Reviewer 3 Report

This article presents findings about the synergistic activity of Ricini semen extract and oxacillin against two SSCmec types of methicillin-resistant Staphylococcus aureus. The same synergistic activity was evaluated for oxacillin and the fatty acids from Ricini semen. The authors propose a possible mechanism of inhibition of Ricini semen combined with oxacillin.

The article is well-written and the results are well-explained. I recommend the publication of the paper with a few minor changes:

Line 92 – Please write “gram-positive” with the capital “G”.

Line 103 – 104 – Could you provide a reference for this?

Line 430 – 438 – The Conclusions section is e bit short. I suggest giving more details about the proposed mechanism of synergistic antibacterial activity

Author Response

  1. Line 92 – Please write “gram-positive” with the capital “G”.

Response: Changed to “Gram-positive” in line 91.

  1. Line 103 – 104 – Could you provide a reference for this?

Response: Added “(data not shown)” in line 103. This is a preliminary result of ours. We do not have a reference.

  1. Line 430 – 438 – The Conclusions section is e bit short. I suggest giving more details about the proposed mechanism of synergistic antibacterial activity

Response: Added detail mechanisms in line 437-440.

 Changed

from “It is suggested that the proposed loss of function of membrane proteins BlaR1 and MecR1, results in suppression of the expression of penicillin-binding proteins 2a and b-lactamase.”

       to “Interfering with the degradation of BlaI and MecI by inactivating BlaR1 and MecR1, respectively, may prevent the expression of b-lactamase (BlaZ) and penicillin-binding proteins 2a (PBP2a), which are resistance mechanisms induced by oxacillin.”

Round 2

Reviewer 2 Report

All comments applied in manuscript file.